# VMseg: Using spatial variance to automatically segment retinal non-perfusion on OCT-angiography

Hugo LE BOITE[1,2]*, Aude COUTURIER[1,2], Ramin TADAYONI[1,2], Mathieu LAMARD[3,4], Gwenolé QUELLEC[3,4]

**1** Université Paris Cité, Paris, France, **2** Ophthalmology Department, AP-HP, Hôpital Lariboisière, Paris, France, **3** Université de Bretagne Occidentale, Brest, France, **4** LaTIM, INSERM UMR 1101, Brest, France

* hugo.leboite@gmail.com

**Data Availability Statement:** Data cannot be shared publicly because they are considered by the RGPD laws in France as biometric data, as they contain fundus images from patients. The SFO

## Abstract

### Background and objectives

To develop and test VMseg, a new image processing algorithm performing automatic segmentation of retinal non-perfusion in widefield OCT-Angiography images, in order to estimate the non-perfusion index in diabetic patients.

### Methods

We included diabetic patients with severe non-proliferative or proliferative diabetic retinopathy. We acquired images using the PlexElite 9000 OCT-A device with a photomontage of 5 images of size 12 x 12 mm. We then developed VMseg, a Python algorithm for non-perfusion detection, which binarizes a variance map calculated through convolution and morphological operations. We used 70% of our data set (development set) to fine-tune the algorithm parameters (convolution and morphological parameters, binarization thresholds) and evaluated the algorithm performance on the remaining 30% (test set). The obtained automatic segmentations were compared to a ground truth corresponding to manual segmentation from a retina expert and the inference processing time was estimated.

### Results

We included 51 eyes of 30 patients (27 severe non-proliferative, 24 proliferative diabetic retinopathy). Using the optimal parameters found on the development set to tune the algorithm, the mean dice for the test set was 0.683 (sd = 0.175). We found a higher dice coefficient for images with a higher area of retinal non-perfusion ($r_s$ = 0.722, p < $10^{-4}$). There was a strong correlation ($r_s$ = 0.877, p < $10^{-4}$) between VMseg estimated non-perfusion indexes and indexes estimated using the ground truth segmentation. The Bland-Altman plot revealed that 3 eyes (5.9%) were significantly under-segmented by VMseg.

ethics committee (contact via ldesjardins@sfo.asso.fr) restricted the full public distribution of the data underlying this study. However, images could be made available on personal request (hugo.leboite@aphp.fr) for academic purposes from research teams, but the sharing of the data would have to be validated by the hospital ethics committee (Lariboisière Hospital, Paris, France) and the SFO ethics committee.

**Funding:** The author(s) received no specific funding for this work.

## Conclusion

We developed VMseg, an automatic algorithm for retinal non-perfusion segmentation on 12 x 12 mm OCT-A widefield photomontages. This simple algorithm was fast at inference time, segmented images in full-resolution and for the OCT-A format, was accurate enough for automatic estimation of retinal non-perfusion index in diabetic patients with diabetic retinopathy.

## 1. Introduction

In diabetic retinopathy, capillary occlusion leads to retinal non-perfusion and ischemia. Like in other diseases associated with retinal ischemia, such as venous occlusion, estimation of retinal non-perfusion area can be used to calculate the retinal non-perfusion index (NPI, corresponding to the surface of non-perfusion divided by the overall surface), which can be used as a marker of severity, or a predictive factor of evolution towards a complication such as proliferative diabetic retinopathy [1]. One of the tools to evaluate NPI is Optic Coherence Tomography–Angiography (OCT-A). This rapid and non-invasive imaging modality uses repeated acquisitions of OCT slices at the same location combined with a decorrelation algorithm to identify vessels as areas where movement of red blood cells has been detected. Using this retinal vessel cartography, non-perfusion can be defined as a surface where retinal vessels and capillaries are absent. OCT-A has demonstrated its capacity to detect retinal non-perfusion and other clinical modifications, in particular in diabetic retinopathy [2–5]. In some settings, OCT-A could even allow retinal non-perfusion visualization better than Fluorescein Angiography, as it can detect capillary networks [6]. However, manual segmentation of all areas of non-perfusion is a slow and difficult process, limiting its use in clinical practice. Automatic OCT-A image analysis has been used with success on small image formats, such as 3 mm x 3 mm and 6 mm x 6 mm, using simple binarization processes to calculate capillary density for example [2, 7–10]. However, using different binarization thresholds and protocols can lead to very different results [11]. Moreover, on larger formats, such as 12 mm x 12 mm, 15 mm x 15 mm or even wider OCT-A image field, the variation of vessel pixel intensity between the center of the image (the macular area) and the periphery does not allow standard binarization algorithms to work properly. Therefore, another approach is needed to automatically analyze widefield images, such as the one proposed in [12], where a semi-automatic protocol using variance maps was used to segment non-perfusion in OCT-A images of size 12 x 12 mm. Non-perfusion areas can be defined as homogeneous (low spatial variance) and dark (low pixel intensity) regions of the image. In this study, we developed VMseg, a non-perfusion segmentation algorithm based on a combination of spatial variance maps, intensity thresholding and morphology operations. We developed and tested VMseg on an even wider OCT-A image field (an automatic photomontage of 5 OCT-A images of size 12 x 12 mm, leading to image sizes of approximately 24 x 24 mm), as retinal ischemia can occur in the periphery and quantifying this peripheral non-perfusion could improve evaluation and prediction of disease severity in diabetic retinopathy.

## 2. Methods

### 2.1 Participants

An observational case series was conducted in a tertiary ophthalmologic center. Treatment-naïve eyes with either severe non-proliferative diabetic retinopathy (NPDR) or proliferative

diabetic retinopathy (PDR) without macular edema were consecutively included over a 6-month period (1st of May 2021–1st of November 2021). We excluded patients with any other retinal disorder (including high myopia) or media opacities such as vitreous hemorrhage or cataract; a history of focal macular laser or PRP, and intravitreal therapy (anti-VEGF or steroids); a history of recent cataract surgery (< 4 months) or pars plana vitrectomy; poor quality of images due to media opacity such as vitreous hemorrhage or cataract. Poor-quality OCT-A images were defined by a signal strength index <6/10 or presence of significant movement or shadow artifact. This study was conducted in accordance with the tenets of the Declaration of Helsinki. All study-related data acquisitions were approved by a French institutional review board (IRB 00008855 Société Française d'Ophtalmologie IRB#1). Verbal consent was obtained from all participants, with information concerning the anonymization process and the possibility of retraction at any time, witnessed by the senior ophthalmologist during the initial consultation and an attending resident. The study did not include minors. No medical intervention was performed and patients had standard-of-care follow-up.

## 2.2 Imaging acquisition and processing

All eyes were imaged using OCT-A (PlexElite OCT-A, Carl Zeiss Meditec, Inc, Humphrey Division, Dublin, California, USA) on an automatic photomontage of 5 images of size 12 x 12 mm (included software module). We exported 2D en-face full-retina slabs (summarizing flow information for the superficial, intermediary and deep retinal capillary plexuses) from the OCT-A device, which we called the "raw" version of the image. For each image, we then used the ImageJ software (https://imagej.net/software/fiji/) to manually crop out areas of low quality, where non-perfusion estimation was difficult, with important movement artifacts (eliminated surface), which we called the "cropped" version of the image. Then, areas of retinal non-perfusion were automatically segmented using a Python script (https://www.python.org/), which will be described in detail in the next section. In order to accelerate manual segmentation, the ground truth was created by correcting the VMseg annotation, eliminating areas that were erroneously segmented, modifying the contours of non-perfusion areas in details if necessary and adding manual segmentation of areas that were not segmented by VMseg. This ground truth manual segmentation was performed by a single trained retina specialist using a contouring tool and a brush tool in ImageJ. We splitted our images into a development set (70% of images), that was used to identify best parameters for the segmentation algorithm, and a test set (30% of images), on which the actual algorithm performance was evaluated.

## 2.3 Automatic segmentation algorithm: VMseg

The first step of the algorithm was to normalize the image and pre-process it using a bilateral filter (opencv library, size of 9 pixels, sigmas values of 40 horizontally and vertically). Then, we created a variance map: the value of each pixel is replaced by the variance of the intensity values of the neighboring pixels. For this purpose, we used a convolution operation with a rectangular kernel of size 3, 5 and 7 pixels, as well as combinations of these 3 kernel sizes: size 3-5-7 for example was created by calculating the mean of the 3 variance maps created using kernel sizes 3, 5 and 7. This parameter was called kernel_size. The obtained variance map was then inverted and intensity was normalized in the range from 0 to 255, so that constant image areas get low pixel values. We attributed a 255 value to pixels with high intensity values in the original image, using a specific threshold (intensity threshold parameter), to ensure that blood vessel pixels get high values. After a second bilateral filtering step, we used morphology operations (closing followed by opening, with a cross-shaped kernel of size 5 pixels) to remove the isolated pixels, considered noise or artifact. The number of morphology operation

iterations was an adjustable parameter. Then, we binarized the modified variance map, using a specific variance threshold. The last step of the algorithm was to find contours of the binary variance map using a contouring algorithm (implemented in Python in the opencv module, called findContours), in order to close areas that presented with holes, and to apply a size threshold of 250 pixels (empirically defined from experience and from the literature [12]) as corresponding to a significantly large non-perfusion area, around 0.15 mm$^2$) and fill the different connected spaces in order to obtain a binary mask of segmentation.

The different steps of the VMseg algorithm are summarized in Fig 1.

## 2.4 Code availability

The Python code for the VMseg algorithm is available on github: https://github.com/HLB32/vmseg.

## 2.5 Performance analysis

In order to evaluate the segmentation performance of the algorithm, we first calculated the dice coefficient, which is defined by the formula below.

$$dice\_coefficient = 2*TP/(2*TP + FP + FN)$$

Where TP is the number of true positive pixels, FP is the number of false positive pixels and FN is the number of false negative pixels. The dice coefficient was calculated for each image, comparing ground truth segmentation to segmentations obtained with VMseg. We tested VMseg on both "raw"images and"cropped" images.

We also evaluated the correlation between NPI calculated from VMseg segmentations (estimated NPI) and NPI calculated from ground truth segmentations (ground truth NPI). NPI

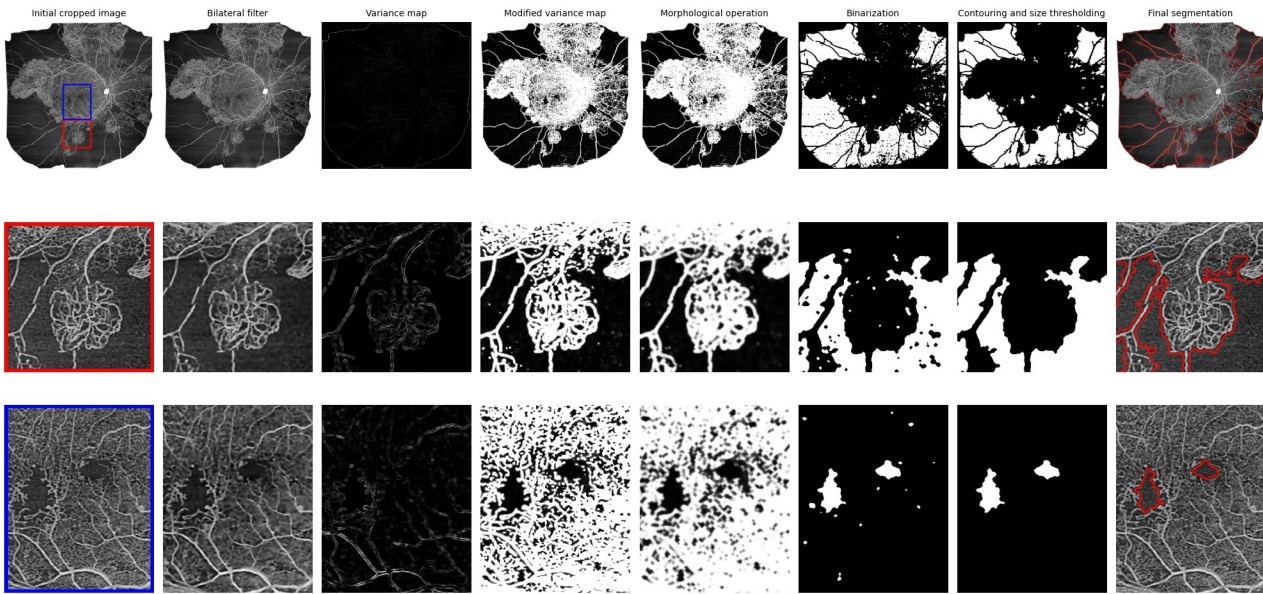

**Fig 1. Summarizing the process of image modification in the VMseg algorithm.** The first row corresponds to the full size OCT-A image. The second and third rows correspond to zoomed patches from the original image. The columns represent the different steps in the VMseg algorithm: bilateral filtering, spatial variance map estimation, variance modification for visualization purposes, morphological operations, binarisation, contouring and size thresholding.

was calculated using the formula below.

$$Non - perfusion\ Index = \frac{Area_{non-perfusion}}{Area_{total} - Area_{eliminated}}$$

Statistical analyses were performed using the R software (https://www.r-project.org/) or Python (https://www.python.org/). Correlation between quantitative variables was evaluated using the Spearman correlation coefficient calculation. Statistical results are shown as mean +/- standard deviation.

For each image, we performed automatic segmentation of non-perfusion with VMseg 10 times in order to estimate the processing time on a computer with a 64 bits 12th Gen Intel(R) Core(TM) i9-12900F 2.40 GHz processor with 32 Go of RAM.

## 2.6 VMseg parameters optimization

We used the development set to identify 3 optimal parameters: the intensity threshold, from a range of 50 to 100 by steps of 1; the variance threshold, from a range of 10 to 30 by steps of 0.5; and the number of morphology operation iterations, from a range of 1 to 4 by steps of 1. For each possible parameter value, we evaluated the dice coefficients on the development set (comparing the obtained segmentation maps to the ground truth segmentations) and kept the parameter value rendering the best mean dice coefficient.

In order to evaluate the robustness of the parameter optimization process, we simulated 10 different random smaller development sets created using 70% of the original development set. On these random development sets, we performed the same optimization process, estimating the best parameter values, and we compared the results to the optimal parameter values obtained on the original development set.

## 3. Results

A total of 51 eyes of 30 patients were included. Patients' mean age was 49.67 ± 14.12 years (range: 24–70) and 21 patients (70%) were male. We included 27 eyes with severe NPDR and 24 eyes with PDR. All patients underwent the same imaging protocol described above.

## 3.1 VMseg parameter optimization

The best performing kernel size combination was: kernel_size = {3, 5}. Both sizes were used and the resulting variance maps were averaged. For the other parameters, as shown in Fig 2, we obtained the best dice coefficients when using a variance threshold of 17 (from a range of 10 to 30 by steps of 0.5) for the variance map binarization, an intensity threshold of 75 (from a

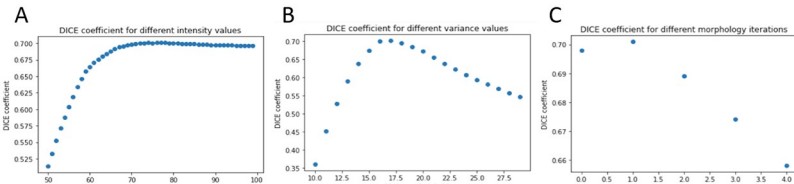

**Fig 2. Fine-tuning VMseg parameters using the development set.** A: Dice coefficient as a function of the intensity threshold. The optimal value was 75, with a slow descending plateau above. B: Dice coefficient as a function of the variance threshold. The optimal variance threshold was 17. C: Dice coefficient as a function of the number of morphology operations on the modified variance map. The optimal number was 1.

range of 50 to 100 by steps of 1) for the pixel intensity normalized image and a number of morphology iterations of 1 (from a range of 0 to 4 by steps of 1).

The analysis of optimization robustness on 10 random smaller development sets created from the original development set showed that the mean optimal intensity threshold was 76.5 +/- 3.34, the mean variance threshold was 17.1 +/- 0.57 and the mean number of morphology operation iterations was 1 (with std = 0, all optimal values being 1). The mean dice coefficients obtained for each random simulated development set at each tested parameter value are shown in Fig 3.

### 3.2 VMseg performances

An illustration of the segmentation results obtained manually and with VMseg is shown in Fig 4. The mean dice coefficients for the raw images were of 0.447 +/- 0.208 and 0.407 +/- 0.177 for the development and the test sets respectively. The mean dice coefficients for the cropped images were of 0.701 +/- 0.194 and 0.683 +/- 0.175 for the development and the test sets respectively. The distribution of dice coefficient values is shown in Fig 5A. There were no statistically significant differences between the dice coefficients in the development and test sets, either for raw images (p = 0.51) or cropped images (p = 0.76). As shown in Fig 5B, there was a positive correlation between dice coefficient and NPI, with the best dice coefficients obtained for images with a higher NPI: the Spearman correlation coefficient was 0.722 with a p-value $< 10^{-4}$.

### 3.3 Non-perfusion index

When comparing ground truth NPI measurements (from the manual segmentation) to estimated NPI (from VMseg automatic segmentation on cropped images), we found a strong and significant correlation (Spearman correlation coefficient of 0.877, p-value $< 10^{-4}$), as shown in Fig 5C. Using a Bland-Altman plot, we found that only 3 images (0.06%) where underestimated beyond the 1.96*standard deviation limit, as shown in Fig 5D.

### 3.4 Processing times

The mean processing time for VMseg to segment the full resolution (1900 by 1900 pixels) OCT-A photomontage was 0.489 seconds +/- 0.043. There was no correlation between mean processing time and estimated NPI (Spearman correlation coefficient of 0.174, p = 0.22).

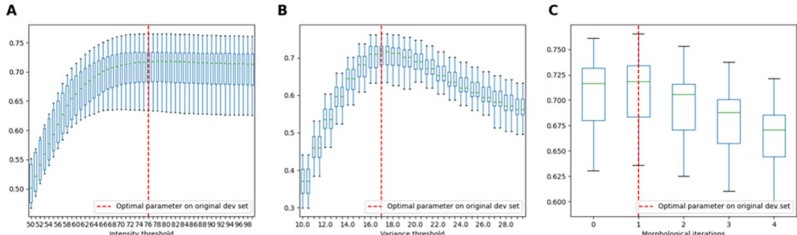

**Fig 3. Robustness of VMseg parameters optimization process.** This figure represents the distribution of mean dice coefficients on 10 random development sets created from the original development set, when testing 3 parameters from the VMseg algorithm. A: Intensity threshold, with a range of 50 to 100, with steps of 1. B: Variance threshold, with a range of 10 to 30, with steps of 0.5. C: Morphology operation iterations number, with a range of 0 to 4, with steps of 1.

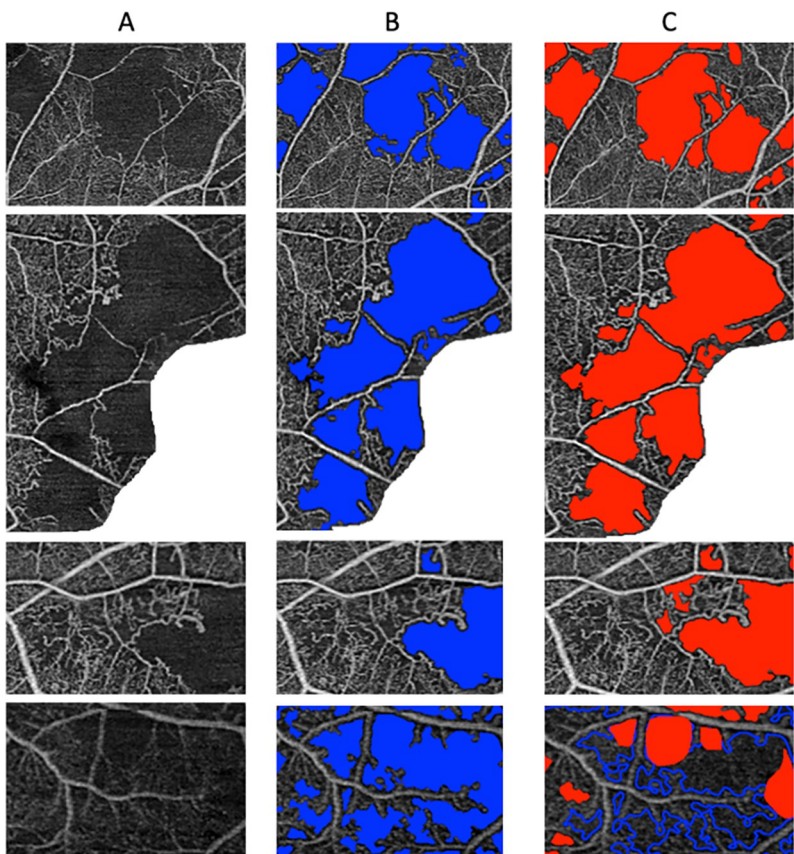

**Fig 4. Illustration of the results of the automatic VMseg algorithm segmentation and ground truth.** A: Original cropped image. The white pixels correspond to areas of the image eliminated because of low image quality. B: Result of the automatic segmentation obtained with VMseg, using optimal parameters found on the development set. C: Result of the manually corrected segmentation mask, considered the ground truth for the algorithm performance evaluation.

## 4. Discussion

We have shown in this study that VMseg, a spatial variance based algorithm, can be used to segment retinal non-perfusion on OCT-A 12 x 12 mm photomontages of diabetic patients with good performances compared to manual segmentation from a retina specialist. Using a development set allowed us to define optimal parameters for the VMseg algorithm, without the need to define subjectively parameters such as binarization thresholds or number of morphology operations. Moreover, with the analysis of robustness of parameters optimization, using a cross validation, we showed that optimal parameters obtained on random simulated development sets were very close to the values obtained on the original development set. This demonstrates that the parameters of the VMseg algorithm are not strongly dependent on the development set used to fine tune the parameters. This reinforces the idea that using a simple spatial variance algorithm like VMseg renders low capabilities to overfit to training data.

The dice coefficients obtained from the cropped images were correct using VMseg, and the processing time was short (around 0.5 second for one image), even though the VMseg algorithm was used on full resolution images (1900 by 1900 pixels). Moreover, we showed that the NPI estimated by VMseg was strongly correlated to the ground truth NPI, which means that VMseg could be used to confidently estimate NPI from OCT-A images in clinical practice, or

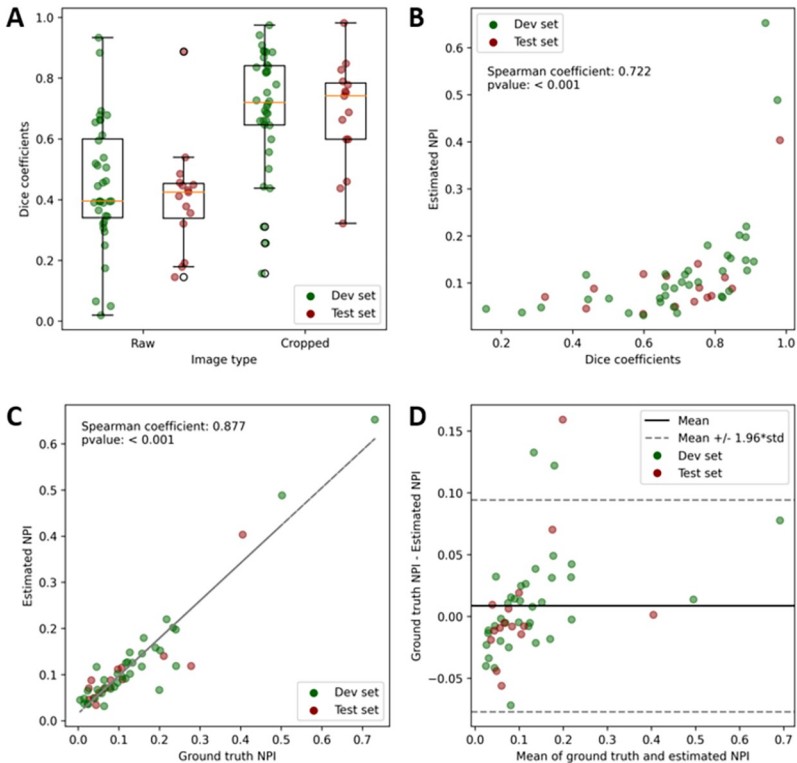

**Fig 5. Comparison between ground truth and VMseg segmentations.** A: Boxplot of the distribution of dice coefficients for the development (green) and test (red) sets, comparing ground truth (manual segmentation) to both raw and cropped images segmented with the VMseg algorithm. B: Correlation between dice coefficients on cropped images and estimated non-perfusion index (NPI). There was a positive correlation, with a Spearman correlation coefficient of 0.722 (p < 0.001), with stronger dice coefficients found in images with higher NPI. C: Strong correlation between ground truth and estimated NPIs, with a Spearman correlation coefficient of 0.877 (p < 0.001). D: Bland-Altman plot, showing on the x-axis the mean of ground truth and estimated NPis and the y-axis the difference of NPIs. We found that only 3 images had an error over the 1.96*standard deviation limit, and these NPI were underestimated by VMseg compared to the ground truth segmentation map.

to feed NPI as a feature to an Artificial Intelligence (AI) model, among other clinical features. As we have shown in a previous work [13], NPI measured on widefield OCT-A could be an important biomarker for proliferative diabetic retinopathy risk estimation.

The method we developed was based on the calculation of variance maps [12]. However, as no details of the algorithm used were available in [12], we could not precisely compare VMseg to the previous work. By looking at the few images available in the original Alibhai et al article, we could however see that predicted non-perfusion areas had sometimes a grainy aspect, with many empty connected spaces within it. This type of artifact was mainly suppressed from the VMseg algorithm by using morphology operations. Further comparative analysis could not be performed.

Our study has several limitations and biases. One bias is that the ground truth segmentation was created using the VMseg segmentation as a base for correction, in order to accelerate the creation of the manual segmentations. Even though the correction was done in details by a retina specialist, correcting the contours of annotated areas if necessary, adding or eliminating areas when segmentation was erroneous, this could introduce a bias in the dice coefficient calculation, as ground truth segmentation could be artificially close to the VMseg segmentation. However, we tried to limit this bias by performing a detailed manual segmentation of the ground truth.

Concerning limitations, first, the parameters identified during the fine-tuning process could be linked to the image format (12 x 12 mm photomontage) and the type of diseases represented in the development set (diabetic retinopathy). The VMseg algorithm might have lower performances when used on other image formats or other diseases (veinous occlusion for example). As we always know the image format when performing the OCT-A image acquisition, we could improve the algorithm by finding optimal parameters for different image formats and using specific parameters at inference time. A way to improve the segmentation performances across different diseases would be to use a deep learning model with a diverse training set. But this approach would not resolve the image format problem, as deep learning models are very sensitive to image format and resolution. Moreover, standard deep learning models for image segmentation, such as U-Nets, do not use the full resolution image at inference time, which could lead to a less precise segmentation prediction. Deep learning models usually have a longer processing time at inference. Furthermore, a larger number of images are needed to train a deep learning model, compared to only a few images to define parameters for a standard image-processing algorithm like VMseg.

Second, even though we found good dice coefficients when comparing ground truth to VMseg on cropped images, we found low dice coefficients when comparing ground truth to VMseg on raw images, which could limit the application of VMseg as a stand alone tool on raw OCT-A images: VMseg should be used on cropped images, which requires a manual segmentation step, or the training of a deep learning model to automatically crop out low quality areas. VMseg could also be used as a tool to diminish the burden of manual annotation of large OCT-A images, in a framework where a retina specialist would correct the VMseg results instead of creating a segmentation map from scratch. However, we did not perform annotation time analysis to demonstrate that VMseg followed by manual correction was significantly faster than manual segmentation from scratch.

Third, we did not compare VMseg to deep learning based models, as we had a limited number of images. However, a deep learning based approach could render better segmentation performances. On smaller OCT-A formats, Giarratano et al [14] have shown that deep learning models, such as standard Convolution Neural Network or U-Nets, give better vessel segmentation results than automatic image processing, using Gabor or Frangi filters for example. In 2018, Guo et al [15] trained MedNet, a U-Net model, on only 180 eyes, to segment retinal non-perfusion on 6 x 6 mm OCT-A images, obtaining dice coefficients ranging from 0.81 for severe diabetic retinopathy to 0.91 for healthy controls. In the future, we could use VMseg to accelerate the segmentation of a large number of OCT-A images, that would be secondarily used as a training set for a deep learning model, after manual verification of the segmentation maps.

## 5. Conclusion

We developed VMseg, an automatic algorithm for retinal non-perfusion segmentation on 12 x 12 mm OCT-A widefield photomontages. We demonstrated in this study that VMseg was fast at inference time (low computational resources), segmented images in full-resolution and for the OCT-A format studied, was accurate enough for automatic estimation of retinal non-perfusion index in diabetic patients with diabetic retinopathy.

## Acknowledgments

We thank Zeiss Inc for having made available to us the use of the Wide-Field Swept-source OCT-A device (PlexElite OCT-A, Carl Zeiss Meditec, Inc, Humphrey Division, Dublin, California, USA).

## Author Contributions

**Conceptualization:** Hugo LE BOITE, Aude COUTURIER, Ramin TADAYONI, Mathieu LAMARD, Gwenolé QUELLEC.

**Data curation:** Hugo LE BOITE.

**Formal analysis:** Hugo LE BOITE.

**Methodology:** Hugo LE BOITE, Aude COUTURIER, Ramin TADAYONI, Mathieu LAMARD, Gwenolé QUELLEC.

**Software:** Hugo LE BOITE, Mathieu LAMARD, Gwenolé QUELLEC.

**Supervision:** Hugo LE BOITE, Aude COUTURIER, Ramin TADAYONI, Mathieu LAMARD, Gwenolé QUELLEC.

**Validation:** Mathieu LAMARD, Gwenolé QUELLEC.

**Writing – original draft:** Hugo LE BOITE, Mathieu LAMARD.

**Writing – review & editing:** Hugo LE BOITE, Aude COUTURIER, Ramin TADAYONI, Mathieu LAMARD, Gwenolé QUELLEC.

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
