## [Decision Letter · Decision Letter 0]

28 May 2024

PONE-D-24-11616VMseg: using spatial variance to automatically segment retinal non-perfusion on OCT-angiographyPLOS ONE

Dear Dr. LE BOITE,

Thank you for submitting your manuscript to PLOS ONE. After careful consideration, we feel that it has merit but does not fully meet PLOS ONE’s publication criteria as it currently stands. Therefore, we invite you to submit a revised version of the manuscript that addresses the points raised during the review process.

We look forward to receiving your revised manuscript.

Kind regards,

Tatsuya Inoue

Academic Editor

PLOS ONE

Journal Requirements:

Reviewers' comments:

Reviewer's Responses to Questions

**Comments to the Author**

1. Is the manuscript technically sound, and do the data support the conclusions?

Reviewer #1: Yes

Reviewer #2: Yes

2. Has the statistical analysis been performed appropriately and rigorously? 

Reviewer #1: No

Reviewer #2: Yes

3. Have the authors made all data underlying the findings in their manuscript fully available?

Reviewer #1: No

Reviewer #2: Yes

4. Is the manuscript presented in an intelligible fashion and written in standard English?

Reviewer #1: Yes

Reviewer #2: Yes

5. Review Comments to the Author

Reviewer #1: This is a significant study and a well-written manuscript in the face of many simple deep learning studies.

Is the reason for not releasing the data because the ethics committee banned it? Please state the reason.

The dice coefficient is not considered to be a figure to be included in the abstract, as the dice coefficient increases if the proportion of areas to be inferred is high.

It is not surprising that the number of cases that fall outside the 95% prediction interval is around 5%. What is important here is the width of the 95% prediction interval.

NPAs are more likely to develop NV when they exceed 30 disk areas, but what is the sensitivity specificity when the threthold is 30 disk areas?

Of the four images in Figure 4, the second image from the top has been cropped, and I would like an explanation of this.

Reviewer #2: The authors present VMseg, a novel algorithm for automatic segmentation of retinal non-perfusion in widefield OCT-Angiography images. The study effectively demonstrates VMseg's capability to estimate the non-perfusion index in diabetic patients, achieving a mean dice coefficient of 0.683 on a test set of 51 eyes from 30 patients. The correlation between VMseg results and manual expert segmentation is strong, indicating the algorithm's accuracy. Additionally, VMseg's fast processing time and robustness make it a promising tool for clinical application in diabetic retinopathy management. The result is interesting and manuscript is well written. I commend the authors for conducting this study. I have the following comments:

Line 84-87 It would be easier to understand if there was more explanation about the background of NPA identification using variance maps.

Line 127-131 Since the ground truth was created using the VMseg annotation, it would be better to have two examiners check the consistency rate.

6. PLOS authors have the option to publish the peer review history of their article (what does this mean?). If published, this will include your full peer review and any attached files.

Reviewer #1: **Yes: **Hidenori Takahashi

Reviewer #2: No

---

## [Author Response · Author response to Decision Letter 0]

11 Jun 2024

5. Review Comments to the Author

Reviewer #1: 

This is a significant study and a well-written manuscript in the face of many simple deep learning studies.

=> Thank you for this kind comment.

Is the reason for not releasing the data because the ethics committee banned it? Please state the reason.

=> Thank you for this question. Yes, the ethics committee banned it and restricted us from releasing the data. 

The dice coefficient is not considered to be a figure to be included in the abstract, as the dice coefficient increases if the proportion of areas to be inferred is high.

=> Thank you for this remark. However, we find that the Dice coefficient is considered to be a correct metric to estimate segmentation performances in various medical image segmentation tasks, and we therefore though it was of interest to present the values we found in the abstract, giving the ability to the reader to get a rapid estimate of the method performance in this specific segmentation task. In our specific case, we found that the Dice coefficient was correlated to the non-perfusion area, but this is not always the case for segmentation models, and we therefore thought it was complementary information and deserved a place in the abstract as well.

It is not surprising that the number of cases that fall outside the 95% prediction interval is around 5%. What is important here is the width of the 95% prediction interval.

=> Thank you for this interesting remark. It is indeed expected from the definition. Given the aspect of the Bland Altman plot, we considered it of interest to describe these 3 images as “significantly under-estimated” as we considered them to be outliers in the distribution, illustrating situations where VMseg significantly underperformed. Taking into consideration your remark, we have enriched the paragraph from the results describing the Bland-Altman plot: the spread of points was correct, without any tendency to under or overestimate the metric, but 3 images could be considered as outliers and were significantly underestimated. 

NPAs are more likely to develop NV when they exceed 30 disk areas, but what is the sensitivity specificity when the threthold is 30 disk areas?

=> Thank you for this interesting question. However, we feel that this question is outside the scope of our study, as we did not study the correlation between the size of non-perfusion areas and the development of new-vessels. It is a very interesting topic, and further investigations in this direction should be carried. 

Of the four images in Figure 4, the second image from the top has been cropped, and I would like an explanation of this.

=> This cropping is described in the methods: it corresponds to a manual step where low-quality areas from the original image were cropped out. This process has been performed on all original images, and in Figure 4, all images are zoomed in, and the second image from the top happens to be at the border of the image, and a cropped out low-quality area is visible.

Reviewer #2: 

The authors present VMseg, a novel algorithm for automatic segmentation of retinal non-perfusion in widefield OCT-Angiography images. The study effectively demonstrates VMseg's capability to estimate the non-perfusion index in diabetic patients, achieving a mean dice coefficient of 0.683 on a test set of 51 eyes from 30 patients. The correlation between VMseg results and manual expert segmentation is strong, indicating the algorithm's accuracy. Additionally, VMseg's fast processing time and robustness make it a promising tool for clinical application in diabetic retinopathy management. The result is interesting and manuscript is well written. 

=> Thank you for this very kind comment.

I commend the authors for conducting this study. I have the following comments:

Line 84-87 It would be easier to understand if there was more explanation about the background of NPA identification using variance maps.

=> Thank you for this remark. We have enriched the corresponding section in order to make it easier to understand this part of the algorithm.

Line 127-131 Since the ground truth was created using the VMseg annotation, it would be better to have two examiners check the consistency rate.

=> Thank you for this remark. It is true that VMseg annotations were used as a base for the manual annotation of ground truth segmentation maps, however this process was meticulous and many small segmentation details were corrected. The use of VMseg as a base was justified by the necessity to make the manual annotation process faster. But this manual annotation and correction was still a very tedious work, and we did not have the resources to create two distinct annotations from two trained retina specialists. In future studies, we are constructing larger and more robust datasets for model training and testing, but these are not applicable to this precise study. We, unfortunately, currently do not have the resources to produce a second annotation for this dataset.

---

## [Decision Letter · Decision Letter 1]

25 Jun 2024

VMseg: using spatial variance to automatically segment retinal non-perfusion on OCT-angiography

PONE-D-24-11616R1

Dear Dr. LE BOITE,

We’re pleased to inform you that your manuscript has been judged scientifically suitable for publication and will be formally accepted for publication once it meets all outstanding technical requirements.

Kind regards,

Tatsuya Inoue

Academic Editor

PLOS ONE

Additional Editor Comments (optional):

Reviewers' comments:

Reviewer's Responses to Questions

**Comments to the Author**

1. If the authors have adequately addressed your comments raised in a previous round of review and you feel that this manuscript is now acceptable for publication, you may indicate that here to bypass the “Comments to the Author” section, enter your conflict of interest statement in the “Confidential to Editor” section, and submit your "Accept" recommendation.

Reviewer #2: All comments have been addressed

2. Is the manuscript technically sound, and do the data support the conclusions?

Reviewer #2: Yes

3. Has the statistical analysis been performed appropriately and rigorously? 

Reviewer #2: Yes

4. Have the authors made all data underlying the findings in their manuscript fully available?

Reviewer #2: Yes

5. Is the manuscript presented in an intelligible fashion and written in standard English?

Reviewer #2: Yes

6. Review Comments to the Author

Reviewer #2: All comments had been adrressed. We appreciate the effort you have put into your research and the thoroughness of your revisions.

7. PLOS authors have the option to publish the peer review history of their article (what does this mean?). If published, this will include your full peer review and any attached files.

Reviewer #2: No

---

## [Editor Report · Acceptance letter]

1 Jul 2024

PONE-D-24-11616R1 

PLOS ONE

Dear Dr. LE BOITE, 

I'm pleased to inform you that your manuscript has been deemed suitable for publication in PLOS ONE. Congratulations! Your manuscript is now being handed over to our production team.

Kind regards, 

on behalf of

Dr. Tatsuya Inoue 

Academic Editor

PLOS ONE